# Immunomodulators Containing Epicor, Colostrum, Vitamin D, Zinc, *Lactobacilli* and *Bifidobacterium* Reduce Respiratory Exacerbations in Children and Adults with Chronic Pulmonary Diseases

Snezhina Lazova [1,2], Nikolay Yanev [3], Nadia Kolarova-Yaneva [4] and Tsvetelina Velikova [5,6,*]

[1] Healthcare Department, Faculty of Public Health, Medical University of Sofia, ul. "Byalo more" 8, 1527 Sofia, Bulgaria
[2] Pediatric Department, University Emergency Hospital (UMHATEM) "N. I. Pirogov", bul. "General Eduard I. Totleben" 21, 1606 Sofia, Bulgaria
[3] Department of Pulmonary Diseases, Medical Faculty, Medical University of Sofia, ul. "Zdrave" 2, 1431 Sofia, Bulgaria
[4] Pediatric Department, Medical Faculty, Medical University of Sofia, ul. "Zdrave" 2, 1431 Sofia, Bulgaria
[5] Clinical Immunology, University Hospital Lozenetz, 1407 Sofia, Bulgaria
[6] Medical Faculty, Sofia University St. Kliment Ohridski, 1407 Sofia, Bulgaria
* Correspondence: tsvelikova@medfac.mu-sofia.bg

**Abstract:** (1) Background: A number of studies have demonstrated the connection between developing or exacerbating chronic respiratory diseases in adults and children. However, still, few studies focus on reducing exacerbations via immunomodulation. (2) Methods: In this pilot study, a total of 25 pediatric and adult patients with bronchial asthma (BA) and chronic obstructive pulmonary disease (COPD)/persistent bacterial bronchitis (PBB) were included, administered over-the-counter (OTC) immunomodulators and followed up for 6 or 12 months. (3) Results: We observed a decrease in the frequency of exacerbations with slight improvements in functional respiratory indicators in adults on their second and third visits and a reduced number of exacerbations and improved spirometry indices in children with BA, although exacerbations requiring hospital admission remained at a similar rate. (4) Conclusions: We confirmed that the number of exacerbations of underlying chronic respiratory disease in adults and children could be reduced after the administration of OTC immunomodulators, probably by optimizing the immune resistance to common viral infections.

**Keywords:** immunomodulator; EpiCor; chronic pulmonary diseases; bronchial asthma; childhood asthma; vitamin D; *S. salivarius*; exacerbations; chronic obstructive pulmonary disease; persistent bacterial bronchitis

## 1. Introduction

Asthma and chronic obstructive pulmonary disease (COPD) are the two most common chronic respiratory diseases, according to the WHO. They affect millions of people all over the world. Exacerbations of both diseases are still a significant public health problem in children and adults [1].

In children, asthma is one of the most common lung diseases, with a frequency varying between 8.3% and 12.3% in industrialized countries. In 80% of cases, the first symptoms debut before age 6 [2], and usually, an allergic aspect of the disease can be proven [3]. The most common symptom for which parents of preschool children visit the nursery is cough. A cough lasting more than 4 weeks is designated as chronic in childhood [4]. Chronic cough can be due to a number of diseases, such as bronchial asthma, upper respiratory tract infection (URTI) and protracted bacterial bronchitis (PBB), as well as chronic suppurative diseases of the respiratory system, such as bronchiectasis [5]. Furthermore, Severe Acute Respiratory Syndrome Coronavirus 2 (SARS-CoV-2), responsible for coronavirus disease

2019 (COVID-19) and considered a global health threat [6], is also connected to a higher risk of infection in patients affected by asthma [7], which is tremendously important during the ongoing COVID-19 pandemic. We also have to emphasize that the pandemic also adversely affects the follow-up of children and adults with chronic obstructive pulmonary disease due to the limited capacity of healthcare systems.

Asthma is characterized by chronic inflammation of the respiratory tract, in which the pathogenesis of inflammatory and structural cells play a role—T-lymphocytes, IgE-producing plasmacytes, eosinophils, mast cells, macrophages, epithelial cells, fibroblasts and smooth muscle cells of the bronchi, as well as an extensive array of proinflammatory and cytotoxic mediators and cytokines (IL-6, IL-8, IL-12, IL-4, IL-10, IL-13, IFN-$\gamma$ and IL-17) [8,9]. This makes it a complex and heterogeneous disease, which is characterized by intermittent and reversible obstruction and chronic inflammation of the respiratory tract due to bronchial hyperreactivity and the infiltration of the respiratory submucosa with immunocompetent cells [3].

Therefore, infections and exacerbations of the underlying disease are expected in children. Similarly, patients with chronic obstructive diseases usually experience secondary immune deficiencies without being diagnosed. It was confirmed that a reduced antiviral immune response might predispose individuals to asthma after a respiratory viral infection [10]. In line with this, since interferons (IFNs, IFN-$\alpha$, INF-$\beta$, IFN-$\gamma$ and IFN-$\lambda$) are essential in the human Th1 immune response to viral infections, respiratory viruses increase the regulation of IFN-responsive genes during infection. Studies have shown that reducing the production of IFN correlates with whistling and asthma following viral infections [11] and may contribute to their exacerbations.

Other studies, however, demonstrated that an allergic Th2 immune response could contribute to the development of asthma by increasing the risk for the disorder [12]. In addition, toll-like receptors also play a role, as well as the change in the restoration and remodeling of the respiratory tract after respiratory viral infection.

Children with PBB usually do not have immunodeficiency; they have age-normal serum levels of immunoglobulins (IgG, IgA, IgM and IgE) and a normal antibody response to proteins (tetanus) and protein–polysaccharide conjugates (e.g., normal vaccine response to H. influenzae type b) [13,14]. On the other hand, intense neutrophilic inflammation in children with PBB was demonstrated in the airway wall with 25.5% to 44% neutrophil involvement [15,16]. Furthermore, elevated levels of interleukin (IL)-8 and IL-1b, which correlate with the degree of neutrophilia, have also been demonstrated [17].

Elevated levels of specific lymphocyte subpopulations—CD56+ and CD16+, known as NK cells—were found, possibly due to an association with accompanying viral co-infections. In addition, the activation of the caspase-1-dependent proinflammatory pathway in response to viruses has been demonstrated in some studies in children with PBB [16].

Therefore, reducing the number and severity of exacerbations also depends on a properly working immune system that can help patients with chronic pulmonary diseases to avoid infections. Furthermore, it is well accepted that children's immune systems are functionally immature, especially during early life. For example, antibodies' capacity for their own production is achieved at 11 years of age [18].

Immunomodulation is a widely accepted approach to optimizing immune protection against various infections and overcoming the typical secondary immune deficiency that accompanies the underlying chronic pulmonary disease in adults and children. The role of immunomodulators without a prescription (i.e., over-the-counter, OTC), especially for patients with chronic conditions, has been studied intensively in recent years [19,20].

It is even more crucial for chronic pulmonary diseases to ensure a properly working immune system and compensate for the secondary immune deficiency that may accompany chronic conditions. In line with these factors, a modulation that reduces viral infections may reduce exacerbations in these patients [21,22].

In this observational study, we aimed to evaluate the impact of immunomodulators containing probiotics, such as *S. salivarius*, Epicor, colostrum, vitamin D, zinc, etc., on the

exacerbations in children with asthma and PBB and adults with bronchial asthma (BA) and COPD in a period of 6 months for the children and 1 year for the adults.

## 2. Materials and Methods

### 2.1. Design of the Study

In this observational study, adult and pediatric patients with a proven diagnosis of chronic pulmonary disease were administered immunomodulatory therapy and evaluated in the 6th and 12th months (for adults) and the 3rd and 6th months (for children) for exacerbation of the primary pulmonary disease (both groups) and the number of hospitalizations (for children).

### 2.2. Subjects of the Study

We investigated a total of 25 patients—10 adults and 15 children with chronic obstructive pulmonary diseases. The group of adult patients consisted of 10 patients (3 with BA and 7 with COPD), followed up for one year. The descriptive characteristics of the adult patients were 4 female and 6 male patients with BA and COPD with an average age of $47 \pm 15$ years. The adult patients were on standard therapy and regimens for BA and COPD, according to GINA and GOLD recommendations [23,24], which were not changed during the follow-up period.

Fifteen children were also enrolled in the study, of whom ten (6–11 years old, four girls and six boys) were diagnosed with BA and five (2–5 years old, two girls and three boys) had recurrent episodes of protracted wet cough (clinically suspected protracted bacterial bronchitis (PBB) or Upper Airway Cough Syndrome (UACS)), as defined by de Benedictis et al. [25]. For better clarity, we defined the first group as the pediatric asthma group and the second as the pediatric wet cough group. The children with BA were on a daily treatment with ICS in a low to medium dose. All of the included asthmatic children were atopic, as their parents and medical records declared. The children with wet cough were on symptomatic treatment when exacerbated during intercurrent viral infections. If wet coughing persisted for more than four weeks, conventional antibiotic treatment was administrated. As an exacerbation in this group, we define a daily productive wet cough for more than two weeks, provoked by intercurrent respiratory viral infections. The pediatric wet cough group included non-atopic children. Blood tests excluded the possible atopy by measuring total IgE, ECP and pediatric-panel-specific aero- and food allergens prior to enrollment.

For the adult subgroup, complete data for forced expiratory volume in 1 s (FEV1), the Asthma Control Questionnaire (ACQ) score for asthma patients and a history of exacerbations were assessed. For the asthmatic children, an initial forced expiratory maneuver, the validated Bulgarian translation of the ACQ and a history of exacerbations were included. For children aged 6–10 years, an interview-based version of the ACQ was used. In addition, historical data for symptom exacerbations and clinical investigation were performed in the preschooler group with protracted wet cough.

Three visits were planned for both adults and children. For adults: initial evaluation (for anamnesis of exacerbations in the previous 12 months), a second visit in the 6th month and a third visit in the 12th month. For the children: initial evaluation (for history data of exacerbations in the previous 12 months), a second visit in the 3rd month and a third visit in the 6th month. Functional testing for adults and children with BA was performed at each visit, focusing on exacerbations (number of exacerbations divided by the number of months between visits).

At the first visit, after the initial evaluation of the patients, they were administered OTC immunomodulators based on the following scheme:

- Adults—three cycles of 2 pills daily for 30 days and a pause of 15 days between them;
- Children—three cycles of 10 mL per os per day for 30 days and a pause of 15 days between them.

The study subjects did not receive any other immunomodulatory therapy.

*2.3. Immunomodulator Content*

The pills for adults contained the following formulation: EpiCor (250 mg), standardized extract of Shiitake (50 mg), colostrum (bovine) (250 mg), andrographolide (30 mg), standardized extract of ginseng (40 mg), vitamin D3 (200 IU, 5 mcg), zinc (7.5 mg), Lactobacillus acidophilus LA85 ($2.5 \times 10^9$ CFU*) and *Bifidobacterium* lactis BLa80 ($2.5 \times 10^9$ CFU*) in two pills as a daily dose.

Children were administered syrup containing EpiCor (250 mg), colostrum (bovine) (200 mg), fructooligosaccharides (100 mg), vitamin D3 (400 IU, 10 mcg), Lactobacillus acidophilus LA85 ($2.5 \times 10^9$ CFU*), *Bifidobacterium* lactis BLa80 ($2.5 \times 10^9$ CFU*) and *Bifidobacterium* infantis BI45 ($1.0 \times 10^9$ CFU) in a 10 mL daily dose.

*2.4. Spirometry*

Lung function testing was performed by conventional spirometry with a Masterscreen Pneumo spirometer '98 (CareFusion Germany, Hoechberg, Germany) before and after administration of a short-acting bronchodilator (Salbutamol) according to the ERS/ATS quality and reproducibility criteria [26,27]. An Easy One Plus Diagnostic spirometer (ndd Medical Technologies®, Andover, MA USA) was used to perform acceptable and repeatable initial spirometry in all asthmatic children, displaying stimulating animation and plotting curves in real time. Individual disposable mouthpiece (spirettes) was used for each child/adult. Children older than four in the chronic wet cough group performed a spirometry trial, which did not reach the ERS/ATS criteria for preschoolers [28]. Given this premise, this result was not included in the study.

*2.5. Ethics*

The study was conducted in accordance with the Declaration of Helsinki. The study was conducted in accordance with the Declaration of Helsinki and ethical review and approval for including children with BA (No. 54/29.06.2015) and adults (No. 72/2019). Formal appropriate clearance was obtained from the ethics committee according to the rules, as this was a pilot study. All included subjects were informed about the OTC immunomodulator and consented to take the pills/syrup. All parents signed informed consent forms for the inclusion of their children, and age-appropriate consent was also obtained from the children themselves. All adults signed informed consent forms before they participated in the study.

*2.6. Statistical Methods*

Statistical analysis was performed with a software package for statistical analysis (SPSS®), IBM 2009, version 19 (2010), and Excel (v. 2010). A significance level of $p < 0.05$ was chosen for the rejection of the null hypothesis.

**3. Results**

*3.1. Adult Patients*

In the studied group of patients, a decrease in the frequency of exacerbations was reported while maintaining functional indicators (Figure 1). There was no significant change in the parameters of the functional respiratory test; the patients remained stable in these parameters. However, we observed a slight improvement (Figure 1).

We found a reduced number of exacerbations on the second and third visits for both adult BA and COPD patients (Figure 2). However, these observations did not reach statistical significance due to the limited number of patients studied.

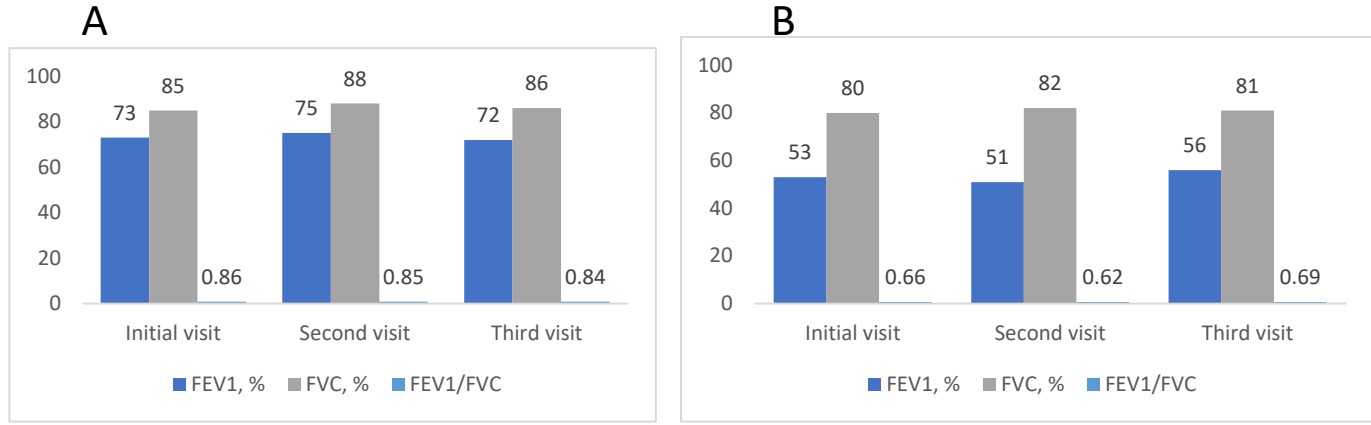

**Figure 1.** Functional indicators (FEV1, %; FVC, %; and FEV1/FVC) in adult patients with asthma (**A**) and COPD (**B**) were assessed on three visits after initiation of immunomodulator administration on the first visit.

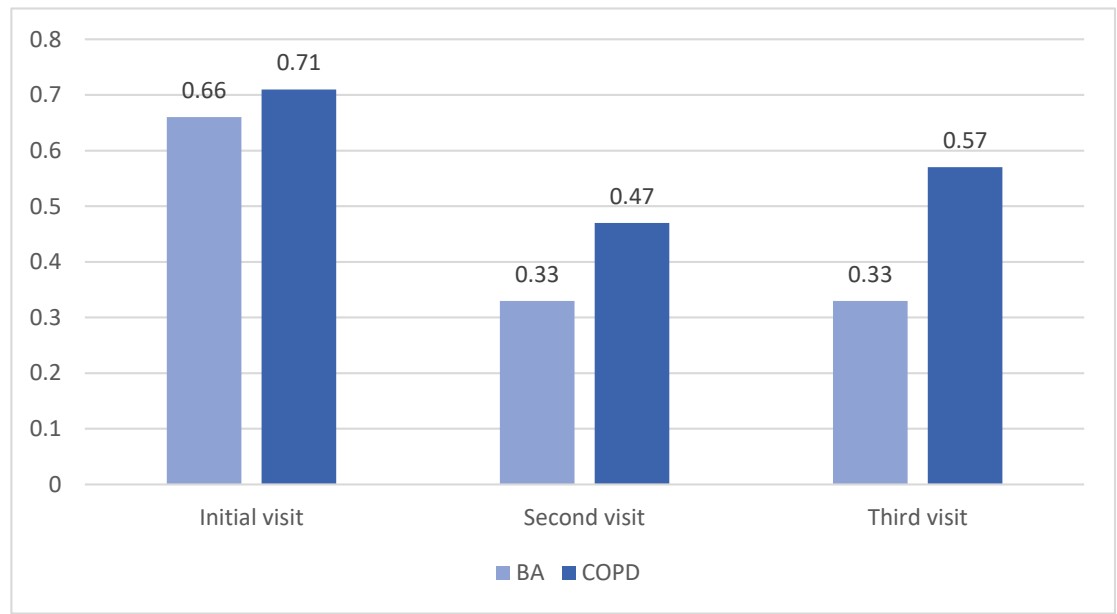

**Figure 2.** Decreased number of exacerbations in adult patients with BA and COPD during the follow-up period after initiation of immunomodulator administration on the first visit.

Nevertheless, we found a trend toward reducing asthma and COPD exacerbations over the one-year follow-up period following the administration of an immunomodulator.

### 3.2. Pediatric Patients

We observed improvements in the spirometry indices of all children on visit 3 (Figure 3).

Similarly, we found a gradually decreased exacerbation frequency in asthmatic children and those with protracted wet cough on visits 2 and 3 (Figure 4). However, our observations were statistically insignificant due to the small number of children.

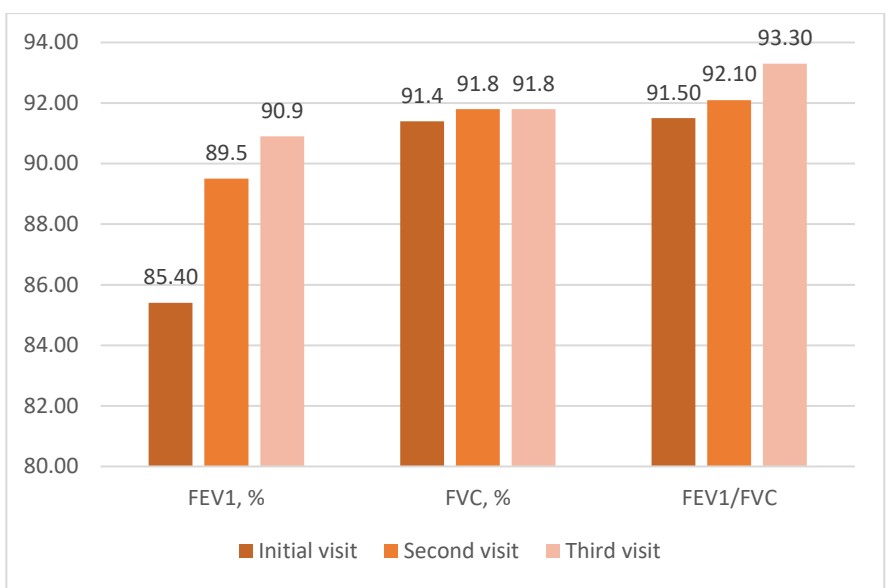

**Figure 3.** Functional indicators (FEV1, %; FVC, %; and FEV1/FVC) in children with asthma were assessed on three visits after immunomodulator administration on the first visit.

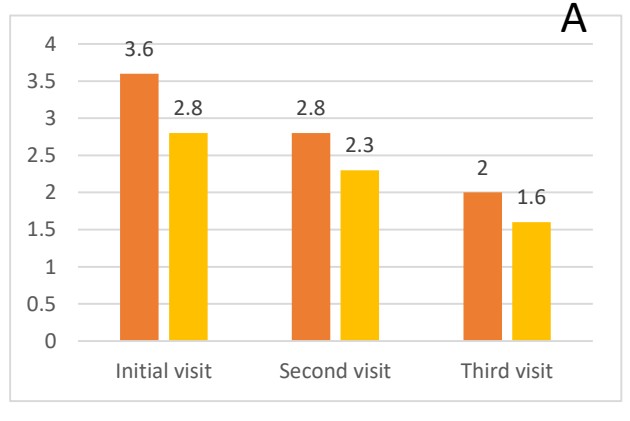

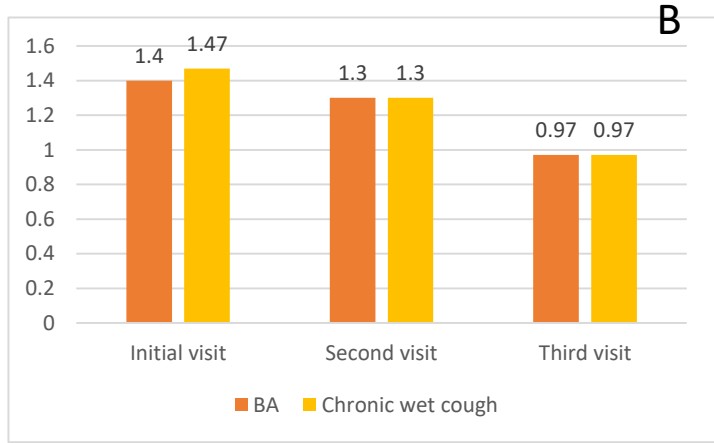

**Figure 4.** Decreased number of exacerbations (**A**) and hospitalizations (**B**) of children with BA and chronic wet cough during the follow-up period after initiation of immunomodulator administration on the first visit.

However, contrary to the adult group of patients' outcomes, we found a more prominent reduction in exacerbations but not hospitalizations for asthmatics. Exacerbations requiring hospital admission remained at a similar rate. In the group of preschoolers, we observed a reduced number of episodes of wet cough exacerbation, usually associated with intercurrent viral infections (Figure 4).

## 4. Discussion

In our observational study, we aimed to assess immunomodulation's effects on episodes of viral-induced exacerbations of asthma and COPD in adults, asthma in school-age children and protracted wet cough in preschoolers. Unfortunately, there is a lack of such studies involving over-the-counter (OTC) immunomodulators with the mentioned composition, and this is a pilot study. Nevertheless, the results are promising, and further placebo-controlled studies with a more significant number of patients and a longer duration of clinical observation are needed.

In this pilot study, we found that in adults, a decrease in the frequency of exacerbations was reported while maintaining functional indicators without changes in the spirometry

indices. In children with BA, we observed a reduction in the exacerbation rate and a slight improvement in lung function. During the follow-up period, there was no change in asthma-related hospitalizations. Similarly, in the preschool group, we observed a reduction in the episodes of protracted wet cough associated with intercurrent viral infections.

In line with this, respiratory viral infections are the most common pathogens associated with wheezing, the exacerbation of chronic obstructive lung disease, and lower respiratory tract infection. Moreover, bacteria may also adhere to host cells through biofilm formation by inducing persistent infections and increasing the risk of developing systemic infections, as a recent observational study demonstrated [29]. Therefore, in line with the increasing antibiotic resistance worldwide, the informed and proper use of antibiotic compounds is required to reduce the risk of resistant bacteria being selected, along with surface functionalization and conditioning.

The appearance and persistence of wheezing at a young age are indicators of developing bronchial asthma. Studies have shown that reducing IFN production and IL-15 deficiency correlates with wheezing episodes and asthma exacerbations that follow viral infections. Allergic Th2 immune responses can also promote the development of asthma after respiratory viral infections. Based on all of these factors, respiratory viruses not only can cause asthma exacerbations but also are associated with an increased risk of developing asthma [30,31].

A recent study found more than one pathogen in the bronchoalveolar lavage (BAL) of children with protracted bacterial bronchitis (PBB). It is suggested that airway inflammation is caused not by a single dominant pathogen but rather by its interaction with an altered microbiota composition (e.g., Prevotella predominance) [32]. A previous viral infection may be a predisposing factor for both bacterial surface attachment and biofilm formation. Viral infections can also be a trigger for an exacerbation of PBB, in which a "planktonic" (free) microorganism is released and grows from the biofilm, generating an increase in the inflammatory response [33,34].

It is well known that viruses can increase the risk of asthma by several pathways. On the one hand, they may increase the production of remodeling factors such as amphiregulin, activist A and vascular endothelial growth factor (VEGF) [31]. These effects can lead to changes in the smooth muscles of the respiratory tract. On the other hand, RSV may increase the production of IL-4 and stimulate the Th2 immune response [12].

Human rhinovirus (HRV) is the most common virus isolated in acute asthma exacerbations, which is observed year-round but is most common in spring and autumn. Influenza viruses (IFVs) and RSV are also important pathogens that are limited to the winter and early spring months. RSV has been linked to children hospitalized with bronchiolitis and adults over 65 [18].

During acute respiratory viral infections, asthma patients have more severe and prolonged symptoms associated with the lower respiratory tract and a stronger descent in respiratory function compared to non-asthmatic patients. The mechanisms that lead to these processes are under-studied. It is assumed that the replication of HRV in bronchial epithelial cells is expressed in patients with bronchial asthma but not in controls.

Another mechanism contributing to exacerbations is IFN response deficiency during viral infection [35]. The interaction of viruses and atopy is well described. Viruses worsen the secretion of IL-8 and TNF from alveolar macrophages after stimulation with two bacterial components, lipopolysaccharides and lipoteichoic acid. Thus, the virus-induced deterioration of antibacterial protection can easily lead to bacterial superinfection. Therefore, pneumococcal vaccination is recommended in patients with asthma over the age of 19 years.

Our previous research found an increased percentage of Th17 cells in children with chronic pulmonary diseases, such as BA [9]. Therefore, we can speculate that the increased percentages of these cells can also lead to frequent exacerbations of the underlying disease.

Furthermore, we discussed in detail that immunomodulators registered as OTC can be used in pediatric and adult patients with chronic pulmonary diseases [18]. Therefore,

we hypothesized that OTC products that provide solid support for immune functions to prevent or attenuate viral infections could serve to decrease the exacerbation of chronic pulmonary diseases.

It is well known that the vast majority of asthma exacerbations in children are infection-induced, mostly virus-induced. Therefore, we can speculate that immunomodulation and reduced intercurrent viral infections can potentially reduce asthma exacerbations. Our six-month observation of 10 asthmatic children aged 6 to 12 years showed a reduction in exacerbations but not in the hospitalization rate. A longer follow-up of immunomodulator administration and a more extensive study group could highlight whether there is a significant improvement. Alongside the exacerbation reduction, we observed improvement in the main spirometry indices.

Moreover, in the study group of preschoolers with recurrent episodes of protracted wet cough, we observed fewer symptom exacerbations on visit 2, which was more convincing on visit 3. However, the limited number of observed children obscures the statistical significance. Further studies could reveal more profound immunomodulation effects on the viral infection frequency and episodes of protracted wet cough in preschoolers.

Based on their content, we chose these two formulations of OTC immunomodulators for adults and children. EpiCor, Shiitake, colostrum and andrographolide, for example, increase NK cell activity in the second hour after administration. Additionally, these natural substances are associated with increased IgA, decreased IgE production, the activation of macrophages and Th cells, and anti-inflammatory activities via blocking NFkB [36–41]. Furthermore, ginseng is associated with rapid IFN I synthesis, which is associated with the effective protection of host cells surrounding the infected cells in the organism [42].

Beta-1,3/1,6-glucans from Saccharomyces cerevisiae are the main ingredients of EpiCor. The activity of EpiCor was demonstrated in a clinical trial assessing the activation of NK cells in the second hour after supplementation [36], while it increased IgA and reduced IgE, again confirmed in a clinical trial [37].

Colostrum is also validated for use as an immunomodulator, solely or in a combination, since it is rich in substances that support immune system functions (i.e., proline, oligosaccharides, lactalbumin, lactoferrin, vitamin D binding protein, vitamins A, B1, B2, B5, B6, B9, B12, C and E, beta-carotene, retinoic acid, sulfur, sodium, chromium, zinc, magnesium, calcium, iron, phosphorus, potassium and IgG, IgA and IgM, in a total of 46.7% proteins, 27.2% carbohydrates and 18% fats) [43]. The vitamin D–binding protein promotes the activation of macrophages and the transport of active metabolites of vitamin D to the site of the local immune response. Colostrum also contains growth factors (such as insulin-like growth factor (IGF-1), transforming growth factors (TGFα and TGFb), epidermal growth factor (EGF), platelet growth factor (PDGF) and growth hormone (GH)). It is also demonstrated that the rich composition of colostrum enhances the immune system by several mechanisms [44].

Along with other functions, lactoferrin has proven antiviral activity through the inhibition of the entry of certain viruses into cells. The other components, such as vitamin D3 and zinc, are also associated with immune protection. Vitamin D increases the synthesis of antimicrobial peptides and the maturation of naïve T cells while reducing the production of proinflammatory cytokines, which may lead to tissue damage. On the other hand, zinc supports the maturation of T and B cells, serves as an antioxidant and also reduces proinflammatory responses [45–47].

Probiotics are also an essential player in supplementation to support the immune system. Recently, we demonstrated that a synbiotic blend containing *Lactobacilli*, colostrum and larch arabinogalactan leads to prominent NK activation in the peripheral blood, providing antiviral activities. At the same time, immunotolerance in the gut is preserved [48,49]. Furthermore, the probiotic blends in the immunomodulators chosen for the current study, *L. acidophilus* and *B. lactis*, were proven as activators of regulatory T cells and NK cells; they also suppress pathogenic bacteria and increase secretory IgA levels.

Thus, in our small study, we confirmed that the administration of specific OTC immunomodulators reduced the number of exacerbations of underlying chronic respiratory diseases (BA or COPD in adults and BA and PBB in children), and we can speculate that this is achieved by optimizing the immune resistance to common viral infections.

## 5. Conclusions

Considering that viral infections may worsen the course of chronic pulmonary diseases in adults and children, we administered OTC immunomodulators with specific contents and followed two groups of patients. Our pilot study confirmed that the number of exacerbations of underlying chronic respiratory diseases (BA or COPD in adults and BA and PBB in children) could be reduced after the administration of OTC immunomodulators, probably by optimizing the immune resistance to common viral infections. However, extensive studies are needed to validate these results. Further, molecular studies may be performed to elucidate the cellular mechanisms of action of immunomodulators in chronic lung diseases.

**Author Contributions:** Conceptualization, S.L., N.Y. and T.V.; data curation, S.L., N.Y. and N.K.-Y.; formal analysis, S.L., N.Y. and N.K.-Y.; investigation, S.L. and N.Y.; methodology, S.L., N.Y. and T.V.; project administration, N.K.-Y. and T.V.; resources, N.K.-Y.; software, T.V.; supervision, T.V.; validation, T.V.; visualization, T.V.; writing—original draft, S.L. and T.V.; writing—review and editing, N.Y., N.K.-Y. and T.V. All authors have read and agreed to the published version of the manuscript.

**Funding:** This research received no external funding.

**Institutional Review Board Statement:** The study was conducted in accordance with the Declaration of Helsinki and ethical review and approval for including children with BA (No. 54/29.06.2015) and adults (No. 72/2019). Formal appropriate clearance was obtained from the ethics committee according to the rules, as this was a pilot study. All included subjects were informed about the OTC immunomodulator (pills/syrup). Informed consent and age-appropriate assent were obtained from all subjects (all parents signed informed consent forms for the inclusion of their children, and age-appropriate assent was also obtained from the children themselves; all adult participants signed informed consent forms).

**Informed Consent Statement:** Informed consent was obtained from all subjects involved in the study.

**Data Availability Statement:** Not applicable.

**Conflicts of Interest:** The authors declare no conflict of interest.

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
