# Peer review of "Immunomodulators Containing Epicor, Colostrum, Vitamin D, Zinc, Lactobacilli and Bifidobacterium Reduce Respiratory Exacerbations in Children and Adults with Chronic Pulmonary Diseases"

_2673-351X, doi:10.3390/sinusitis6020009_

Round 1

Reviewer 1 Report

Review report

Brief Summary

The objective was to study the effects of immunomodulator "Over The counter" OTC drugs on chronic obstructive pulmonary diseases in both Children and adults.  This was a pilot study with a smaller sample size of 10 adults and 15 children.  The adults were followed up to 1 year and children up to 6 months. The episodes of exacerbations were reduced in adults and children, on the other hand, showed better improvement of Functional indicators and decreased exacerbations.

My Specific comments/ Suggestions for improvement

Section 2.4 Ethics Line 174: “approval was waived for this study” – approval could NOT have been waived as this study involves special groups - children and involves OTC drugs, please clarify to? “formal appropriate clearance was obtained from the ethics committee according to rules as this was a pilot study”.

Line 176: “All  parents signed their informed consent for  the inclusion of their children.” Please add that age-appropriate assent was also obtained from the children like this. “All  parents signed their informed consent for  the inclusion of their children and age-appropriate assent was also obtained from the children themselves.”

Section Institutional Review Board Statement Line 370: “approval was waived for this study” - approval could NOT have been waived as this study involves special groups - children and involves OTC drugs, please clarify to? “formal appropriate clearance was obtained from ethics committee according to rules as this was a pilot study”.

Line 346: “administatrriotn” spelling mistake change to “administration”

Line 372: “they consent to take the pills/syrop”- please rewrite to “they gave written consent/ age-appropriate assent to participate in the study”

Line 373 Informed Consent Statement: “Informed consent was obtained from all subjects” change to? “Informed consent and age-appropriate assent was obtained from all subjects”

Line 350 Conclusions  I suggest adding the line? “Further, molecular studies may be performed to elucidate the cellular mechanisms of action of immunomodulators in chronic lung diseases”

The article is very well drafted and contains plenty of defense for their synthesis.  Most references are the latest, and the authors have perused and included the latest evidence in their synthesis. The language and style of the paper are also flawless and praiseworthy.

Author Response

Review report

Brief Summary

The objective was to study the effects of immunomodulator "Over The counter" OTC drugs on chronic obstructive pulmonary diseases in both Children and adults.  This was a pilot study with a smaller sample size of 10 adults and 15 children.  The adults were followed up to 1 year and children up to 6 months. The episodes of exacerbations were reduced in adults and children, on the other hand, showed better improvement of Functional indicators and decreased exacerbations.

My Specific comments/ Suggestions for improvement

Section 2.4 Ethics Line 174: “approval was waived for this study” – approval could NOT have been waived as this study involves special groups - children and involves OTC drugs, please clarify to? “formal appropriate clearance was obtained from the ethics committee according to rules as this was a pilot study”.

  • Thank you very much for the critical point. We agree with the referee, and also, we were approached by the editorial office of the journal, thus, we provided the ethical review and approval for including subjects – both children and adults. And also, we would like to confirm that all subjects were informed about the design signed informed consent to be included in the study.

Line 176: “All  parents signed their informed consent for  the inclusion of their children.” Please add that age-appropriate assent was also obtained from the children like this. “All  parents signed their informed consent for  the inclusion of their children and age-appropriate assent was also obtained from the children themselves.”

  • We completely agree with the note, we have added this statement in the text and at the end of the paper.

Section Institutional Review Board Statement Line 370: “approval was waived for this study” - approval could NOT have been waived as this study involves special groups - children and involves OTC drugs, please clarify to? “formal appropriate clearance was obtained from ethics committee according to rules as this was a pilot study”.

  • The referee is right to point this out, and we add this statement in the ethics of the paper – within the text and at the end.

Line 346: “administatrriotn” spelling mistake change to “administration”

  • Thank you for noticing that mistake, we corrected it.

Line 372: “they consent to take the pills/syrop”- please rewrite to “they gave written consent/ age-appropriate assent to participate in the study”

  • Thank you for the clarification, we have corrected the text in the recommended manner.

Line 373 Informed Consent Statement: “Informed consent was obtained from all subjects” change to? “Informed consent and age-appropriate assent was obtained from all subjects”

  • Thank you very much for all the suggestions to refine the Ethics of the study and to improve the quality of the paper.

Line 350 Conclusions  I suggest adding the line? “Further, molecular studies may be performed to elucidate the cellular mechanisms of action of immunomodulators in chronic lung diseases”

  • Thank you for the valuable suggestion. We added this passage in the conclusion.

The article is very well drafted and contains plenty of defense for their synthesis.  Most references are the latest, and the authors have perused and included the latest evidence in their synthesis. The language and style of the paper are also flawless and praiseworthy.

  • Thank you very much for the overall evaluation of our paper as good.

Reviewer 2 Report

The manuscript entitled "Immunomodulators containing Epicor, colostrum, vitamin D, zinc, Lactobacilli and Bifidobacterium reduced the respiratory exacerbations in children and adults with chronic pulmonary diseases" is interesting for idea but it has an important issue related the ethical issue. In fact, in this work, at the first visit, after the initial evaluation of the patients, patients with asthma and COPD   were administered on OTC immunomodulators on the following scheme:  • Adults – three cycles of 2 pills daily for 30 days; and a pause of 15 days between them;  • Children – three cycles of 10 ml per os per day for 30 days, and a pause of 15  days between them. However, the treatment of asthma and COPD should be followed GINA and GOLD recommendations. The treatment with immunomodulators have not been recommended for patients with asthma and COPD.

Author Response

The manuscript entitled "Immunomodulators containing Epicor, colostrum, vitamin D, zinc, Lactobacilli and Bifidobacterium reduced the respiratory exacerbations in children and adults with chronic pulmonary diseases" is interesting for idea but it has an important issue related the ethical issue. In fact, in this work, at the first visit, after the initial evaluation of the patients, patients with asthma and COPD   were administered on OTC immunomodulators on the following scheme:  • Adults – three cycles of 2 pills daily for 30 days; and a pause of 15 days between them;  • Children – three cycles of 10 ml per os per day for 30 days, and a pause of 15  days between them. However, the treatment of asthma and COPD should be followed GINA and GOLD recommendations. The treatment with immunomodulators have not been recommended for patients with asthma and COPD.

  • Thank you very much for the critical and valuable comments regarding our paper. We completely agree with the concerns of the referee regarding the therapy of the patients. We would like to assure the referee and the editorial board that the adult patients were on standard therapy and regimens for BA and COPD, in accordance with GINA and GOLD, which were not changed during the follow-up period. we have stated this in the manuscript (lines 111-112). We completely agree that the OTC immunomodulators are NOT recommended as treatment options, therefore, we add these as supplements, as the included OTC was registered as a supplement. We also mentioned GINA and GOLD in the text, following your suggestion.

Reviewer 3 Report

Comments on Lazova et al:

The aim of this manuscript is to evaluate the impact of immunomodulators, containing probiotics, on the exacerbations of children and PBB, and adults with BA and COPD, in a timeline of six months, as regards the children and one year, for the adults.

This manuscript shows rich content, providing a deep insight for some works: the study is within the journal’s scope, and I found it to be well-written, providing sufficient information. Even if the manuscript provides an organic overview, with a densely organized structure and based on well-synthetized evidence, there are some suggestions necessary to make the article complete and fully readable. For these reasons, the manuscript requires major changes.

Please find below an enumerated list of comments on my review of the manuscript:

INTRODUCTION:

LINE 41: Please, reformulate this sentence. “The most common symptom, for which parents of preschool children visit the nursery, is cough”.

LINE 46: Furthermore, Severe Acute Respiratory Syndrome Coronavirus 2 (Sars – CoV – 2), responsible of Coronavirus disease 2019 (COVID – 19) and considered a global health threat (see, for reference: Torge, D.; Bernardi, S.; Arcangeli, M.; Bianchi, S. Histopathological Features of SARS-CoV-2 in Extrapulmonary Organ Infection: A Systematic Review of Literature. Pathogens 202211, 867. https://doi.org/10.3390/pathogens11080867), is also connected to a higher risk of infection, in patients affected by asthma (see, for reference: Palmon PA, Jackson DJ, Denlinger LC. COVID-19 Infections and Asthma. J Allergy Clin Immunol Pract. 2022 Mar;10(3):658-663. doi: 10.1016/j.jaip.2021.10.072. Epub 2021 Nov 25. PMID: 34838708; PMCID: PMC8613003). The manuscript should provide an organic overview of this chronic obstructive pulmonary disease, by also contextualizing it in the light of the ongoing pandemic. This is a significant issue of this manuscripts, which could help to improve its impact.

DISCUSSION:

LINE 259: Moreover, bacteria may also adhere to host cells, through biofilm formation, by inducing persistent infections and also increasing the risk of developing systemic infections (see, for reference: Bernardi, S.; Anderson, A.; Macchiarelli, G.; Hellwig, E.; Cieplik, F.; Vach, K.; Al-Ahmad, A. Subinhibitory Antibiotic Concentrations Enhance Biofilm Formation of Clinical Enterococcus faecalis Isolates. Antibiotics 202110, 874. https://doi.org/10.3390/antibiotics10070874). This observational study should highlight the linkage between viral infection and biofilm formation, by providing recent evidence on this topic.

The main topic is interesting, and certainly of great clinical impact. As regards the originality and strengths of this manuscript, this is a significant contribute to the ongoing research on this topic, as it extends the research field on the impact of immunomodulators, containing probiotics, on the exacerbations of children and PBB, and adults with BA and COPD. Overall, the contents are rich, and the authors also give their deep insight for some works.

As regards the section of methods, there is a specific and detailed explanation for the methods used in this study: this is particularly significant, since the manuscript relies on a multitude of methodological and statistical analysis, to derive its conclusions. The methodology applied is overall correct, the results are reliable and adequately discussed.

The conclusion of this manuscript is perfectly in line with the main purpose of the paper: the authors have designed and conducted the study properly. As regards the conclusions, they are well written and present an adequate balance between the description of previous findings and the results presented by the authors.

Finally, this manuscript also shows a basic structure, properly divided and looks like very informative on this topic. Furthermore, figures and tables are complete, organized in an organic manner and easy to read.

In conclusion, this manuscript is densely presented and well organized, based on well-synthetized evidence. The authors were lucid in their style of writing, making it easy to read and understand the message, portrayed in the manuscript. Besides, the methodology design was appropriately implemented within the study. However, many of the topics are very concisely covered. This manuscript provided a comprehensive analysis of current knowledge in this field. Moreover, this research has futuristic importance and could be potential for future research. However, major concerns of this manuscript are with the introductive and discussive section: for these reasons, I have major comments for these sections, for improvement before acceptance for publication. The article is accurate and provides relevant information on the topic and I have some major points to make, that may help to improve the quality of the current manuscript and maximize its scientific impact. I would accept this manuscript if the comments are addressed properly.

Author Response

Comments on Lazova et al:

The aim of this manuscript is to evaluate the impact of immunomodulators, containing probiotics, on the exacerbations of children and PBB, and adults with BA and COPD, in a timeline of six months, as regards the children and one year, for the adults.

This manuscript shows rich content, providing a deep insight for some works: the study is within the journal’s scope, and I found it to be well-written, providing sufficient information. Even if the manuscript provides an organic overview, with a densely organized structure and based on well-synthetized evidence, there are some suggestions necessary to make the article complete and fully readable. For these reasons, the manuscript requires major changes.

  • Thank you very much for the overall evaluation of our paper as good. We have revised the paper according to all the suggestions and recommendations.

Please find below an enumerated list of comments on my review of the manuscript:

INTRODUCTION:

LINE 41: Please, reformulate this sentence. “The most common symptom, for which parents of preschool children visit the nursery, is cough”.

  • Thank you very much for your recommendation how to improve the clarity of this fragment.

LINE 46: Furthermore, Severe Acute Respiratory Syndrome Coronavirus 2 (Sars – CoV – 2), responsible of Coronavirus disease 2019 (COVID – 19) and considered a global health threat (see, for reference: Torge, D.; Bernardi, S.; Arcangeli, M.; Bianchi, S. Histopathological Features of SARS-CoV-2 in Extrapulmonary Organ Infection: A Systematic Review of Literature. Pathogens 2022, 11, 867. https://doi.org/10.3390/pathogens11080867), is also connected to a higher risk of infection, in patients affected by asthma (see, for reference: Palmon PA, Jackson DJ, Denlinger LC. COVID-19 Infections and Asthma. J Allergy Clin Immunol Pract. 2022 Mar;10(3):658-663. doi: 10.1016/j.jaip.2021.10.072. Epub 2021 Nov 25. PMID: 34838708; PMCID: PMC8613003). The manuscript should provide an organic overview of this chronic obstructive pulmonary disease, by also contextualizing it in the light of the ongoing pandemic. This is a significant issue of this manuscripts, which could help to improve its impact.

  • The referee is right to point out that these facts should be included and cited in the paper in the context of ongoing pandemic.

DISCUSSION:

LINE 259: Moreover, bacteria may also adhere to host cells, through biofilm formation, by inducing persistent infections and also increasing the risk of developing systemic infections (see, for reference: Bernardi, S.; Anderson, A.; Macchiarelli, G.; Hellwig, E.; Cieplik, F.; Vach, K.; Al-Ahmad, A. Subinhibitory Antibiotic Concentrations Enhance Biofilm Formation of Clinical Enterococcus faecalis Isolates. Antibiotics 2021, 10, 874. https://doi.org/10.3390/antibiotics10070874). This observational study should highlight the linkage between viral infection and biofilm formation, by providing recent evidence on this topic.

  • Thank you very much for the valuable suggestion, we have insert this within the manuscript.

The main topic is interesting, and certainly of great clinical impact. As regards the originality and strengths of this manuscript, this is a significant contribute to the ongoing research on this topic, as it extends the research field on the impact of immunomodulators, containing probiotics, on the exacerbations of children and PBB, and adults with BA and COPD. Overall, the contents are rich, and the authors also give their deep insight for some works.

  • Thank you very much for the high assessment.

As regards the section of methods, there is a specific and detailed explanation for the methods used in this study: this is particularly significant, since the manuscript relies on a multitude of methodological and statistical analysis, to derive its conclusions. The methodology applied is overall correct, the results are reliable and adequately discussed.

  • Thank you very much for the honest

The conclusion of this manuscript is perfectly in line with the main purpose of the paper: the authors have designed and conducted the study properly. As regards the conclusions, they are well written and present an adequate balance between the description of previous findings and the results presented by the authors.

Finally, this manuscript also shows a basic structure, properly divided and looks like very informative on this topic. Furthermore, figures and tables are complete, organized in an organic manner and easy to read.

  • Thank you very much for the high assessment.

In conclusion, this manuscript is densely presented and well organized, based on well-synthetized evidence. The authors were lucid in their style of writing, making it easy to read and understand the message, portrayed in the manuscript. Besides, the methodology design was appropriately implemented within the study. However, many of the topics are very concisely covered. This manuscript provided a comprehensive analysis of current knowledge in this field. Moreover, this research has futuristic importance and could be potential for future research. However, major concerns of this manuscript are with the introductive and discussive section: for these reasons, I have major comments for these sections, for improvement before acceptance for publication. The article is accurate and provides relevant information on the topic and I have some major points to make, that may help to improve the quality of the current manuscript and maximize its scientific impact. I would accept this manuscript if the comments are addressed properly.

  • Thank you once again for the critical notes. We agree that we missed some essential topics to discuss in regards with our paper. We have implemented the suggestion and add information about the current pandemic, and also the role of infections in BA development. We believe that this has improved our paper significantly.

Round 2

Reviewer 2 Report

The manuscript has been improved and can be accepted for publication.

Reviewer 3 Report

Authors complied to suggestions.manuscript can be now accepted